



# 1 ISASO2 : Recent trends and regional patterns of Ocean Dissolved Oxygen change

Nicolas Kolodziejczyk[1], Esther Portela[1], Virginie Thierry[1], Annaig Prigent[1]
[1] *Univ. Brest, CNRS, Ifremer, IRD, LOPS laboratory, IUEM, Plouzané, France*
*Correspondence to*: Nicolas Kolodziejczyk (nicolas.kolodziejczyk@univ-brest.fr)
**Abstract.** Recent estimates of the global inventory of dissolved oxygen (DO) have suggested a decrease of 2% since the
1960s. However, due to the sparse historical oxygen data coverage, the DO inventory exhibits large regional uncertainties at
interannual timescale. Using ISASO2, a new DO Argo-based optimally interpolated climatology
https://doi.org/10.17882/52367(Kolodziejczyk et al.,2021), we have estimated an updated regional oxygen inventory. Over
the long term (~1980-2013), comparing the ISASO2 Argo fields with the first guess WOA18 built from the DO bottle
samples fields extracted from WOD18, the broad tendency to global ocean deoxygenation remains robust in the upper 2000
m with -451±243 Tmol per decade. The oxygen decline is more pronounced in the key ventilation areas of the Southern
Ocean and North Atlantic, except in the Nordic Seas, where oxygen has increased. Over the shorter timescale of the Argo
period (2005-2019), the deoxygenation tendency seems globally amplified (-1211±218 Tmol per decade). However, DO
changes exhibit stronger amplitude and contrasted regional patterns, likely driven by interannual modes of variability in
regions as, for instance, the North Atlantic Subpolar-gyre. The recent changes in Apparent Oxygen Utilization mainly
explain the interannual variability in the ventilation regions. However, Argo DO coverage is still incomplete at the global
and calibration method developpement are still in progress. Continuing the monitoring of the seasonal to interannual and
regional to global DO variability from ISASO2 will improve our ability to reduce uncertainties on global and regional DO
inventory.

## 20 1 Introduction

22       The global inventory of dissolved oxygen (DO) has been reported to decline about 2% over the last 60 years
(Schmidtko et al., 2017 ; Helm et al., 2011 ; Ito et al., 2017). Moreover, future projections suggest a sustained-to-increasing
ocean deoxygenation until 2100, depending on the emission's scenario (Bopp et al., 2013). This trend has been partly
explained by the loss of solubility due to the warming of the upper ocean layer under global warming of the Earth system
(Keeling et al., 2010; Schmidtko et al., 2017). Furthermore, ocean warming and melting of continental ice have
consequences on ocean circulation by enhancing surface stratification (Li et al., 2020 ; Yamaguchi & Suga, 2019; Sallée et



al., 2021; Bronselaer et al., 2018; IPCC, 2021), which may reduce the ventilation of the DO in the Southern Ocean (Helm et
al., 2011; Couespel et al., 2019; Bronselaer et al., 2020), in regions of deep water formation in the northern North Atlantic
(Stendardo & Gruber, 2012), and in the shallow thermocline ventilation cells in tropical regions (Oschlies et al., 2018). The
latter has particular impact on the weak DO supply to the Oxygen Minimum Zones (OMZs), that are naturally low-oxygen
regions located in the eastern tropical oceans (Karstensen et al., 2008 ; Paulmier and Ruiz-Pino, 2009, Hahn et al., 2017).
One of the critical manifestations of the global deoxygenation is the expansion of the OMZs (Stramma et al., 2008) with
strong impact on the habitat of pelagic species (Stramma et al., 2012) and macrofaunal diversity (Sperling et al., 2016).
Many efforts have been made recently to gather a comprehensive DO dataset and to diagnose the DO global
inventory from historical data sets (e.g. Schmitdko et al., 2017; Helm et al., 2011 ; Ito et al, 2017). However, the interannual
and regional DO variability and driving mechanisms, especially those associated to the ocean ventilation (Helm et al., 2011;
Portela et al., 2020b), suffer from large uncertainties due to a lack of dedicated and sustained observing systems (Levin,
2018; Oshlies et al., 2018). Consequently, at the regional scales, the modes of natural ocean variability need to be better
identified, as they can obscure the long-term anthropogenic oxygen trends (e.g. Stramma et al., 2020;  Stramma and
Schmidtko, 2021; Feucher et al., 2022).

Since 2005, the development of the BGC-Argo mission collecting biogeochemical parameters such as DO (Claustre
et al., 2009; Roemmich et al., 2019) from Argo floats has already provided more than 150,000 quality controlled profiles
(Thierry and Bittig, 2021; Maurer et al., 2021). Although still sparse at global scale, the coverage of the Argo DO time series
allows resolving the seasonal-to-interannual DO variability at regional scale. In some key regions, such as the North Atlantic
subtropical and subpolar gyres, this new dataset has already improved our understanding of the physical drivers of regional
DO variability (e.g. Billheimer et al., 2021; Feucher et al., 2022). Thus, sustained and consistent further observations are
needed to monitor the oxygen regional and interannual variability and to disentangle the long-term anthropogenic trends
from natural variability (Levin et al., 2017). Also, beyond the warming induced ocean deoxygenation, the crucial role of



ventilation change in the amplification/attenuation of deoxygenation should be addressed at the regional and interannual
timescale.

In this study, we constructed a climatological gridded product of the most updated Argo DO dataset using the In
Situ Analysis System tool (ISAS, Gaillard et al., 2009, 2016). ISAS has been routinely used to optimally interpolate in situ
temperature and salinity data, and it has now been adapted for analysis of the Argo DO. The new ISAS climatologies over
selected periods between 2005-2019 are compared with the historical World Ocean atlas (WOA18, representative of the
early 1980s). The global patterns of the recent DO change over the Argo period (2005-2019) and the long-term variability
(1980-2013) are eventually discussed (in regions covered by Argo DO). The climatological ISASO2 fields are freely
available at: https://doi.org/10.17882/52367.
**2 Data and Method**
**2.1 Argo DO data**
The Argo DO data (2021-03 release) have been downloaded from Coriolis Global Assembly Center (Argo, 2000).
In this study, we used 117,359 Delayed Mode (DM) Argo S-profiles from the surface to 2000 m depth covering the period
2005-2019 (Fig. 1a). Only Quality Control (QC) flags set to 1 and 2 have been retained by default in the analysis process.

There are two main methods to measure DO in the ocean from an autonomous platform (Thierry et al. 2022). The
first method is an electrochemical method that uses a Clark-type polarographic cell (Gnaigner and Foster, 1983). The SBE43
sensor uses this measurement principle. The second method is an optical method based on the principle of dynamic
fluorescence quenching (Lakowicz, 2006). Sensors based on this method are referred to as oxygen optodes (e.g. Aanderaa
4330, SBE63 and SBE83, AROD-FT Rinko). While the two types of sensors have been tested in the field since 2005, it has
been shown that oxygen optodes are the only dissolved oxygen sensors currently suitable for Argo applications (Bittig et al,
2019). In addition, sensor models, designs, calibration processes and computing equations have evolved since 2005 (Thierry
et al. 2022, Bittig et al., 2018, 2019) with the overall aim to improve DO data accuracy.




The best accuracy that can be reached with the present sensor type and knowledge is 1-2 µmol kg$^{-1}$ (0,5% $O_2$
saturation). This requires the use of optodes with individual multipoint calibration and the adjustment of oxygen data to *in*
*situ* and/or in air reference data (Bittig and Körtzinger,. 2017) to correct for the drift in $O_2$ sensitivity that occurs between
calibration and deployment ("storage drift" of order ~5% year$^{-1}$), as well as the "in situ drift" (order of –0.5% year$^{-1}$) that
occurs during the multiyear deployment period (Bittig et al., 2018, 2019). In addition, oxygen optodes can show a pressure
dependent response of the sensor. This pressure has been characterized and is taken into account in the dissolved oxygen
computation from the raw data (Bittig et al., 2015, 2018, Thierry et al., 2022). For some floats, an additional pressure
correction might be necessary to reach the 1-2 µmol kg$^{-1}$. It needs to be estimated from ship-based calibrated reference data
(Racapé et al. 2019). The last issue concerns a time-dependent lag in response to a change in DO, which affects data
accuracy when sensors experience O2 gradients during vertical float displacement. While methods to characterize the
response time and to correct for the sensor lag are known (Bittig et al., 2014; Bittig and Körtzinger, 2017; Gordon et al.,
2020), they cannot be systematically applied as it requires a timing of each observation, which is not available on all float
models. The lack of time response correction can lead to uncertainty of 6-7 µmol kg$^{-1}$ for SBE63 sensor and 13-15 µmol kg$^{-1}$
for Aanderaa 4330 sensor in the most strong oxycline region (Bittig and Kortzinger 2017) and might contribute to the bias of
- 1.18 mmol kg$^{-1}$ observed on the float data compared to GLODAP (Sharp et al., 2023).

The distribution of measurement error provided by Argo DO DMQC reflects the evolution of the sensors and the
associated data uncertainties. The measurement errors, as estimated by DMQC operators, are mainly distributed around 2-3
µmol kg$^{-1}$, 7-8 µmol kg$^{-1}$ and 13-14 µmol kg$^{-1}$ (Fig. 2). The former one corresponds to the mean accuracy expected from the
current sensor type and knowledge (Thierry and Bittig, 2021, Maurer et al., 2021). The two latter ones mainly reflect the data
accuracy of older sensor models, but might be due to specific configurations (deployment in strong oxycline region or lack
of reference data). The ISASO2 climatology will be regularly updated, including the updated DMQC S-profiles with the
most recent advance in sensor corrections.





Another source of uncertainty is the inhomogeneous spatio-temporal distribution of the Argo DO data. Data
coverage remains sparse in subtropical and tropical regions of the South Atlantic and Indian Ocean, as well as in the eastern
subtropical area of the Pacific Ocean (Fig. 1c). The coverage is particularly poor in the Pacific Ocean, while the best sampled
regions are the North Atlantic, the Southern Ocean, the Eastern North Pacific and the North Indian Oceans (Fig. 1c).

**2.2. WOA18 and WOD18 OSD data**
The DO annual climatology from WOA18 (Garcia et al., 2019) was used as a first guess for ISASO2 interpolation.
WOA18 was constructed by objective analysis (Barnes, 1964) of the available historical Ocean Station Data (OSD) provided
within the World Ocean Database, 2018 release (WOD18; Boyer et al., 2019). The OSD WOD18 DO dataset gathers
962,219 low vertical resolution profiles over the period 1899-2017, with a strong sampling bias in the Northern Hemisphere
(Fig. 1b). The OSD-DO data are obtained from the Winkler titration method, excluding other DO sensor devices (Garcia et
al., 2019). The accuracy of the Winker method was established as ±0.3 µmol kg$^{-1}$ (Carpenter, 1965). WOA18 climatology is
provided on a 1° grid with an effective horizontal resolution between 892 km and 446 km, and over 102 standard depth
levels between the surface and 5500 m depth. As there is no preferred time correlation scale introduced in the WOA18
mapping (Garcia et al., 2019), the time representativeness of the WOA18 has been estimated using the OSD profiles
distribution of WOD18 (Fig. 1a).

The bulk of profiles deeper than 1000 m depth was collected mainly over the period 1970-1988 (1st and 3rd quartiles
; Fig. 1a). The spatial distribution of the time representativeness of the WOA18/WOD18 for profiles deeper than at least
1000 m depth (Fig. 3d,e) reveals that the median year for the sampling carried out in the Northern Hemisphere ocean falls
between 1970-1990 with a variance ranging within 6-20 years. In the Southern Hemisphere the sampling is more recent,
between 1980-2000, with the largest spread (up to 25 years) in the South Atlantic Ocean (Fig. 3e). Using these relative
time-representativeness values for computing the decadal trends at each grid point, leads to increased uncertainties in the
estimate of the median time  (see Fig. 3). Over the vertical, the time representativeness of WOA18/WOD18 gives a median



time between 1970-1990 in Northern Hemisphere with deeper profile reaching 2000 m depth, while in the Southern
Hemisphere the median time is more representative of the 1980-2000 period with larger spreads for samples deeper than
1000 m depth (Fig. 4).

**2.3 ISAS**

Using the In Situ Analysis System tool (ISAS; Gaillard et al. 2009, 2016), an Optimal Interpolation (OI, Bretherton
et al., 1976) scheme has been used to map the DO data. The horizontal grid was 0.5° over 187 vertical levels from surface to
5500 m depth. The a-priori statistics used in the ISASO2 (ISAS for O2) optimal Interpolation are the DO annual climatology
from WOA18 (Garcia et al., 2019) (Fig. 5a), used as a first guess, and updated standard deviations (STD) from Argo DO
(binned in 5°x5° boxes ; Fig. 5d). The updated DO STD computed from Argo is representative of the seasonal to interannual
variance. In comparison to WOA variance climatology, new Argo DO data have significantly improved the a priori
estimation of the DO variance in regions historically ill-sampled, such as the Southern Hemisphere (see Fig. 5b,d). New
regions of high DO variability have been identified, such as in the western boundary currents and the Antarctic Circumpolar
Current. These regions are associated with frontal regions with sharp contrasts between water masses and strong dynamical
features that likely produce high DO variability (e.g. Chapman et al., 2020).

The Argo DO sampling coverage is not yet evenly distributed over the global ocean. Therefore, to deal with the
partial coverage, the objective mapping was first performed for the mean state over three selected multiyear periods. In order
to be consistent with the spatial scale of ISAS20 temperature and salinity fields also used in this study (Kolodziejczyk et al.,
2021), the zonal and meridional decorrelation radius and variance weights are the same as in Gaillard et al. (2016; their
equation 4), except for the correlation timescale, the time correlation Gaussian function, defined as:

$$C(dt) = \sigma exp\left(\frac{dt^2}{2L_t^2}\right) (1),$$


where $L_t$ is the time decorrelation scale. The three configurations are: i) ISASO2_MEAN that maps the whole Argo data set
using a time correlation Gaussian window centered on January 2013 with a five-year correlation scale (half Gaussian width



is $L_t$ = 5 years). This mean encompasses 10 years of data and is designed to be representative of the decadal mean over the
period 2009-2018. ii) ISASO2_M11 and ISASO2_M16 are pentadal means using a time correlation Gaussian window
centered on January 2011 and 2016, respectively, with a 2.5 years correlation scale (encompassing five years of data). These
two configurations are designed to be representative of the ocean's pentadal mean over the periods between 2009-2013 and
2014-2018, respectively. This approach was chosen for being a trade-off between having sufficient spatial coverage and
allowing a temporal discretization.

156        In addition, the DO concentration at saturation [$O_2^{sat}$] has been computed using the ISAS20 T/S monthly fields

averaged over the same three periods (Kolodziejczyk et al., 2021). The DO and Apparent Oxygen Utilisation (defined as
AOU= [$O_2^{sat}$]-[$O_2$]) mean climatology have been provided by the World Ocean Atlas 2018 (WOA18, Garcia et al., 2019).

160        It is worth addressing the innovation of the DO Argo profiles using WOA18 first guess in the OI ISASO2

procedure. Clearly, the time representativeness of both datasets is different. The distribution of the difference between DO
Argo observations and WOA18 first guess at the location of the DO Argo profiles shows an average shift of -4.31 µmol.kg$^{-1}$
(Fig. 6). This suggests a general deoxygenated bias between the WOA18 and Argo data likely due to either i) a mean
deoxygenation at the location of the DO Argo profile, or ii) remaining low oxygen bias in DO Argo profiles. In contrast,
after the OI procedure, the median of residuals, i.e. the difference between DO Argo observations and ISASO2_MEAN at
the location of the DO Argo profiles, is -0.39 µmol.kg$^{-1}$ (Fig. 6). This suggests that ISASO2_MEAN estimates are slightly
more oxygenated than DO Argo data at the location of the DO Argo profiles. More generally, the DO ISASO2_MEAN
slightly overestimates the DO in the period 2005-2019. This is likely due to the relaxation of the DO estimates towards the
first guess (WOA18) which is typically less oxygenated, and the smoothing in the ISAS OI procedure.

**2.2. Error handling**





The analyzed variance (error) is provided from the OI procedure. It is derived from the a priori variance updated
from new Argo DO (Fig. 1c), covariance scales and measurement error (Fig. 2) (Gaillard et al., 2009, 2016). Then, the error
estimates of the regional DO inventory is computed by propagating the analyzed error computed from ISASO2 (Fig. 5d),
using the linear expression:
$$\sigma^2 = \sum_{i}^{N} \alpha_i \sigma_i + \sum_{i}^{N} \sum_{j \neq i}^{N} \alpha_i \alpha_j \sigma_i \sigma_j \rho_{ij} \quad (1)$$

where $\alpha_i$ is an element of space (layer thickness multiplied by the mean density for vertical inventory or grid cell surface for
global inventory), $\epsilon_i$ the error at the i grid point, and $\rho_{ij}$ is the correlation between the i and j grid points. This equation was
applied for vertical and horizontal summation of error. Correlation $\rho_{ij}$ was computed from ISAS20 monthly temperature
fields, assuming that large physical correlation scales can be applied to DO profiles. For computing the WOA18 DO
inventory error, the a priori variance is inaccurate and an analyzed error is not provided. Therefore, the full a priory variance
derived from Argo DO was used. This should be considered as an upper bound of the WOA18 DO inventory.

The error on a zonal average section is computed for each section cell using Smith et al. (1994) expression for
averaging spatial error :
$$\epsilon^2 = \frac{\overline{\epsilon_i^2}}{Dof} \quad (2)$$

where, $\epsilon_i^2$ is the quadratic average of the individual analyzed error along a zonal band; and *Dof* is the degree of freedom
corresponding to the number of independent grid cells, *i.e.* the total number of grid cells divided by the number of grid cell
corresponding to the largest correlation scale used in ISAS (Gaillard et al. , 2016; von Schukmann et al., 2009).
**2.3. Computation of trends**
The computation of the decadal (pseudo)trend of the DO inventory or concentration is :



$$\Delta O_2^{trend} = 10 \times \frac{O_2^{clim1} - O_2^{clim2}}{T_1 - T_2} \quad (2)$$

where $O_2^{Clim1}$ and $O_2^{clim2}$ are the DO inventory maps or concentration sections computed from two distinct means
and $T_1$, $T_2$ are their respective median time. In the case of the three ISASO2 means, this is the central time of the analysis. In
the case of the WOA, we have taken the representative time estimated using the median time distribution of the WOD18
OSD profiles (Fig. 3d,e). In both cases, we computed decadal trends, even if the difference between ISASO2 means is
representative of only a five-year variability. This choice has been made to make comparable (normalize) the magnitude of
the long-term and interannual variability. The associated uncertainty ranges for DO inventory and concentration trends have
been estimated using propagation techniques of the analyzed error of ISASO2 (detailed in Supplementary Information). The
calibration biases may affect the long-term trends. Introducing artificial bias correction (around 1 µmol.kg$^{-1}$) does not change
the regional pattern of DO increase/reduction (not shown). However, mean bias smaller than -1 µmol.kg$^{-1}$ could dramatically
impact the total global inventory long-term trend, as our calculation method would overweight the Argo data in the
comparison with WOA climatology.

Three climatological configurations representative of 3 different periods are used to compute DO change, named i)
ISASO2_MEAN minus WOA18, which corresponds on average to the period Argo (2009-2018) minus pre-Argo and ii)
ISASO2_M16 minus ISASO2_M11, which corresponds to the period 2018-2014 minus 2013-2009 (five-year average). The
differences are normalized using the time span between WOA18/WOD18 local median year (Fig. 3d and 4) and 2013
(central time of ISASO2_MEAN), and between 2016-2011 (5 years), respectively. The computed trend is normalized in
terms of equivalent trend in µmol kg$^{-1}$ or Tmol per decade. For both periods, the error is computed by propagating the error
as follows :
$$\epsilon \approx \left| \frac{x_i^a - x_j^a}{T} \right| \sqrt{\frac{\epsilon_i^2 + \epsilon_j^2}{\left(x_i^a - x_j^a\right)^2} + \frac{\epsilon_T^2}{T^2} + cov} \quad (3)$$



where, $x^a_{i,j}$ are the analyzed fields with i,j in WOA, ISASO2_MEAN, ISASO2_M11, and ISASO2_M16 , T is the median
time span between two compared analyzes i,j or estimated by WOD18 profiles median time in the case of using WOA18 (see
map Fig. 1c) , $\epsilon_{i,j}$ error of analysis, $\epsilon_T$ the variance of the median time span between analysis i,j  (map Fig. 3d), *cov* is the
covariance between different errors which is assumed to be zero. For the WOA18 as no analysis error is provided, we have
taken STD computed from DO Argo (Fig. 5d). This estimate is likely an upper bound of the errors of analysis (corresponding
to a percentage of variance (PCTVAR) of 100% in the OI).

In order to assess the robustness of our approach to estimate the change in DO, we have computed the DO trends in
5°x5° boxes using a linear regression of individual profiles at 500 (not shown) and 1000 m depth (± 50 m) using OSD plus
Argo data (Fig. 7c), WOD18 alone (Fig. 7d), and Argo data alone (Fig. 7e). Three features can be highlighted from this
analysis: (i) the comparison with computation of equivalent trends from OSD and ISAS interpolated fields provides the same
order of magnitudes and comparable regional pattern of trends (Fig. 7a,b,c,e). (ii) The most striking features are the larger
magnitudes of the trends (in absolute value) during the Argo period, reaching more than twice those of the pre-Argo period.
(iii), using the OSD dataset alone over the pre-Argo period (Fig. 7d) provides inconsistent trends (compared to Argo+OSD,
Fig. 7c) in ill-sampled regions such as the Southern Ocean, and generally non-significant trends due to lack of data in many
other regions (Fig. 7d).

In order to get further confidence in the mapped trends, we have also computed DO trends at selected locations over
the global ocean (Fig. 8). In historically ill-sampled regions, such as the Southern Ocean, Subtropical and mid-latitude
regions, trends are generally barely significant and not consistent among dataset (Fig. 7 and 8a,d,e). In these cases, Argo
floats provide data over the recent period that allowed to better constrain the long term trend. In contrast, well sampled
regions, such as North Atlantic, North Pacific and, Nordic Seas and, exhibit robust long-term trends (Fig. 7-8b,c,f).





## 3 Results

The updated global DO inventory computed from ISASO2_MEAN gives about 235.2±0.1 Pmol (236±0.1 Pmol for WOA18O2), which is close to the 227.4±1.1 Pmol computed from Schmidko et al. (2017) with a different data set and mapping method. The ~1980-2013 global DO inventory change given by ISASO2_MEAN minus WOA18 is around -451±243 Tmol per decade for the upper 2000 m depth.

The global map of DO inventory change depicted in Figure 9a, reflects the regional mean equivalent trends between the Argo and pre-Argo period. The global ocean has generally deoxygenated over several decades, except in the Nordic Seas (+ 4 mol.m$^{-2}$ per decade) and in few regions of the Subtropical Southern Indo-Pacific (> +10 mol.m$^{-2}$ per decade; Fig. 9a). The Northern Atlantic and Pacific Ocean are deoxygenating at a rate around -6 mol.m$^{-2}$ per decade, and the tropical OMZs in the majority of the basins have expanded at 1000 m depth (black and red contours, Fig. 9a). The addition of new DO Argo data in this study has provided more robust patterns in the mid-latitude Southern Ocean in regions were Argo floats have been deployed (Fig. 9a). The Southern Ocean has been historically poorly sampled in terms of DO, thus its contribution to global deoxygenation since 1980 (about 42%) might have been underestimated or impossible in previous studies (Ito et al., 2017). This DO loss is comparable to that of the Tropical band over the same period (43% between 30°S-N).

The DO Argo data provide further insight on the interannual changes of the DO inventory over the last 10 years (Fig. 9b). Globally, the ocean has lost the equivalent of about -1211±218 Tmol of DO per decade over the Argo period (deduced from ISASO2_16 minus ISASO2_11). This suggests a much more intense interannual variability in comparison to the longer-term change previously reported in the literature and in this study (e.g Schmidtko et al., 2017). Furthermore, over the Argo period, the regional patterns are more than twice as intense as the regional patterns of the long term trends, but in line with regional change due to natural mode of variability (e.g. Stramma and Schmidtko, 2021; Fig. 9ab). The North Atlantic Subpolar Gyre, the Nordic Sea, and the Gulf Stream regions show the largest spatially coherent increase of DO inventory (> + 20 mol.m$^{-2}$ per decade, Fig. 9b). In contrast to the long-term trends, the Southern Ocean around 35-50°S (i.e.



subtropical front) exhibits a more intense increase of DO inventory (> + 10 mol.m$^{-2}$ per decade) while, south of 50°N, more
intense deoxygenation dominates (< -15 mol.m$^{-2}$ per decade).

263          The processes generally invoked to explain the global deoxygenation are the decrease in seawater

solubility ([O2$^{sat}$]) directly induced by the ocean warming, and reduced ocean ventilation combined with increased biological
utilization. These last two processes reflect an increase of the AOU (a proxy of the water-mass age). Zonally averaged
sections of DO concentration, [O2$^{sat}$] and AOU as well as their temporal changes are used to better illustrate and understand
interannual and long term DO variability and their driving mechanisms (Fig. 10 and Fig. 11-14). Long-term regional
deoxygenation (Fig 10d) occurs mainly in the Southern Ocean and Northern Hemisphere, in the upper 1500 m, except in the
subpolar northern latitudes (45-60°N) where it reaches down to 2000 m depth (not enough deeper DO Argo data are
available; Fig. 10ad) with an average trend of around 2 µmol.kg$^{-1}$ per decade. This is mainly explained by deoxygenation in
NADW in the Atlantic basin (Fig. 11ad). In contrast, the water column in the Nordic Seas (>60°N) has oxygenated, mainly
below 1000 m depth (Fig. 10d and 11d). The Southern Ocean has generally deoxygenated between the lower Antarctic
Intermediate Waters and the Upper Circumpolar Deep Waters ($\sigma_\theta$~27.2-27.8 kg.m$^{-3}$, ~2 µmol.m$^{-2}$ per decade, Fig. 10a,d). In
contrast, deepening of the isopycnals of the well ventilated SubAntarctic Mode Water (SAMW), indicates a volume increase
of the water masses, although with no significant DO increase (Fig. 10a,d). Between 20°S-20°N, the tropical band exhibits
slight deoxygenation in the upper 1000 m (1 µmol.kg$^{-1}$ per decade). The tropical and equatorial deoxygenation, mainly
observed in the east of the Atlantic and Pacific and northern Indian basins, is consistent with the observed decadal increase
of the OMZs volume (Stramma et al., 2008; Fig. 10a,b and Fig. 11-13a,d).

280          The observed last-decade change reveals that the interannual-to-decadal variability can be twice as large as the long

term trend in some specific regions (Fig. 11). As an exception, over the Argo period, the Northern Hemisphere north of 40°N
exhibits an oxygenation signal (>5 µmol.m$^{-2}$ per decade Fig. 10g) that opposes the long-term mean, especially in the North
Atlantic Subpolar gyre, down to 1500 m depth (>10 µmol.kg$^{-1}$ per decade, Fig. 11g). In the Southern Hemisphere there are
differences between ocean basins. In the Indo-Pacific basins (Fig. 12g and 13g), over the Argo period, the deoxygenation is



more intense and deeper than in the pre-Argo period. Particularly, the DO loss within the Upper Circumpolar Deep Water
(around $\sigma_\theta$=27.5 kg.m$^{-3}$) exceeds 8 µmol.kg$^{-1}$ per decade (Fig. 10g and Fig 11-14g). In contrast, the Atlantic basin exhibits a
smaller oxygen loss or even oxygen gain over the upper 1000 m, and a slightly more intense deoxygenation between 1000m
and 2000 m (Fig 11d,g).

290       Over the recent (long-term) period, solubility contributed to ~30% (50%) of the global deoxygenation in the upper

2000 m of the water column (Fig.10e,h). Over both periods, the regional (de)oxygenation appears to be mainly driven by
AOU-related processes (Fig. 10f,i and 12-14f,i). AOU changes are also dominant over the shorter time scales, as they
explain the DO change patterns in most regions and depth ranges (Fig. 10i and 12-14i). Although it is complex to disentangle
the physical ventilation from biological consumption in AOU changes, four main patterns show up from the Figure 10f,i
(and Fig. 11-14f,i) : i) AOU Increase in the Southern Ocean below the Intermediate Waters ($\sigma_\theta\geq$27.5 kg.m$^{-3}$); ii) AOU
Increase within the tropical band of the three oceanic basins (Fig. 10f,i and 11-14f,i); iii) AOU increase (decrease) of the
surface to subsurface waters over the long-term period (Argo period) in the North Atlantic Subpolar gyre and North Pacific
(Fig. 11,12f,i); and iv) AOU decrease North of 70°N in the Nordic Seas around 1000 m depth (Fig. 11f).
**DataAvailability** : Data described in this manuscript can be accessed at SEANOE under https://doi.org/10.17882/52367
(Kolodziejczyk et al., 2021)



## 4 Discussion and Conclusion

The updated ISASO2 climatologies, with DMQC DO Argo profiles, provide new insight on the recent change in global and regional patterns of DO in the context of the global ocean deoxygenation. First, comparing the ISASO2 Argo fields with the WOA18 built from the DO bottle samples fields extracted from WOD18, i.e. the most reliable historical dataset, the broad tendency to global ocean deoxygenation remains robust in the upper 2000 m with -451±243 Tmol per decade between 1980-2013. In spite of a large range of values and no common time and space coverage over the estimated values, our results are in line with previous studies. Using WOCE data, Helm et al. (2012) found -550±25 Tmol per decade within 100-1000 m depth surface layer between 1970-1990 ; Schmidko et al., (2017) found a trend of -961 ± 429 Tmol per decade in the full water column between 1960-2015; and Ito et al. (2017) reported DO trends of -243±124 Tmol per decade in the upper 1000 m depth between 1958-2015. In spite of different methods in estimating the uncertainty among the literature, the relative magnitude of uncertainty estimate on global trends is also generally in line with previous studies.

Although DO Argo coverage is still partial in some regions, some contrasted regional patterns between the pre-Argo and Argo periods are emerging from the updated dataset. For instance, the historically ill-sampled Southern Ocean shows clear deoxygenation over at least the last two decades. Moreover, consistent sampling over the Argo period has allowed us to provide pentadal to decadal mean for these regions. Although the deoxygenation tendency seems globally amplified (-1211±218 Tmol per decade), we cannot argue an acceleration of deoxygenation over the Argo period. Rather, the global inventory may be sensitive to very large regional and interannual variability, as suggested by our map and sections of interannual change. As reported in recent studies (Stramma et al., 2020; Stramma and Schmitdko, 2021, Feucher et al., 2022), natural modes of variability are known to strongly impact the ocean DO change at regional scale. The Argo DO standard deviation and trends estimates shown here, reveal for the first time the very inhomogeneous and intense DO variability over the ocean. This may help to quantify the range of uncertainty induced by natural/interannual variability over the longer-term deoxygenation trends. For instance, the North Atlantic Subpolar gyre exhibits a recent oxygenation, explained by AOU decrease due to the intensification and cooling of the subpolar gyre and the return of deep convection favouring the sequestration of DO in the intermediate Labrador Sea Water, as reported in the Irminger Sea (Tjiputra et al.,



2018, Feucher et al., 2022). In this region, the interannual to decadal variability of ventilation makes it difficult to observe a
significant  long-term deoxygenation trend (Feucher et al., 2022).

Noticeable oxygenation patterns are observed in the deep Nordic Seas, especially north of 70°N in the Greenland
Sea. Since the late 90's, deeper convection has been reported in the Greenland Sea (1500 m depth) while the temperature of
intermediate water has increased (Brakstad et al., 2019). The latter is compatible with observed continuous decrease in
oxygen solubility. However, deeper convection contributes to replenish DO at depth, making the intermediate water younger
(decreasing AOU) and then, overcoming the solubility decrease (Lauvset et al., 2018). Interestingly, besides the strong
interannual variability, the long-term DO decline in North Atlantic, both in Labrador Sea Water and NADW, as well as DO
increase in the deep Greenland Sea, suggest a northward shift of the DO ventilation. In climate model future projections, a
warmer and more stratified ocean suggest a northward shift of the AMOC ventilation hot-spots as the winter sea-ice
maximum extension retreats (Lique & Thomas, 2018).

The Southern Ocean, which has been historically less consistently sampled, also reveals the large deoxygenation
that has been reported near the Antarctic continent. The input of  meltwater in Antarctica, and poleward wind shift in the
Southern Ocean have been suspected to be responsible for increasing stratification. The latter would reduce ventilation and
water-mass age (AOU) near the continent and within the Circumpolar Deep Water with a potential strong impact on
biological productivity (Bronsealer et al., 2016, 2018). Interestingly, SAMW that is a hot-spot of DO ventilation (Portela et
al., 2020b) does not show any significant change over the long term in spite of their volume increase during the last decades
(Kolodziejczyk et al., 2019, Portela et al, 2020a). In contrast, over the Argo period, strong variability of
deoxygenation/oxygenation patterns is reported around the Southern Ocean, with oxygenation of the mode/intermediate
waters during the last decade and deoxygenation of Circumpolar Deep Water. In the latter case, wind and isopycnal mixing
change (Abernathy and Ferreira, 2015; Naveira-Garabato et al., 2017), as well as stronger stratification (Bronsealer et al.,
2018) have been suggested to play a key role in modulating tracer ventilation in the Southern Ocean (Morrison et al., 2022).





The precise role of long-term increased stratification and/or change in isopycnal mixing, as well as the recent Southern
Ocean deoxygenation should be addressed in future studies.

Our study is mainly based on the historical bottle samples (OSD WOA18) and the recent DO Argo dataset
(ISASO2). Excluding other oxygen profiles over the historical period may cause less resolved historical time series and
spatial coverage, especially in the poorly sampled regions like the Southern Ocean. This may increase the uncertainties on
trends computation: as it is well known, the lack of data coverage may result in underestimating the global trends computed
from parameters analyzed with optimal interpolation methods (Lymann and Johnson, 2011). However, continuous sampling
of the Argo network helps to remove the seasonal bias usually associated with the historical cruises mainly occurring during
summer. It also allows better estimates of the DO natural seasonal-to-interannual STD which is crucial for computing the
range of uncertainties (Fig. S1d), even if the DO eddy variance contribution is still to be estimated (Atkins et al., 2022).

A major source of uncertainty on DO Argo data in this study is the calibration of the DO profiles, which can exhibit
remaining low oxygen bias larger than 1 to 3 $\mu mol.kg^{-1}$ that could be due to uncorrected sensor time response (Bittig et al.,
2014 ; Maurer et al., 2021, Sharp et al., 2023). Continuous monitoring and progress in Argo DO time response and drift
correction is mandatory to ensure quality and consistency among the DO dataset of different origins. Regarding the historical
OSD data, the accuracy of the Winkler titration method is generally ±0.3 $\mu mol\ kg^{-1}$ (Carpenter, 1965). However, larger
uncertainties are expected due to change of the sampling and titration methods over the last century. Nevertheless, the still
sparse DO data is probably the main source of uncertainties, especially when using the ISAS_M11 and ISAS_M16
configurations that present the most limited Argo coverage of the 5-years windows. Also, we hope that the ongoing
deployment of global Argo DO observations will help to update the ISASO2 analysis in order to provide DO global fields
with better spatial coverage and enhanced time discretization.

Therefore, long-term observations and consistent spatial coverage of the ocean DO, combined with other BGC
parameters in the framework of OneArgo will provide further insight on the regional mechanisms of ocean deoxygenation.



Moreover, this can finally shed some light on their still poorly known oxygen biological drivers (Levin et al., 2017; Oschlies
et al., 2018).
**Competing interests :** The contact author has declared that none of the authors has any competing interests.
**Acknowledgment** : ISAS tools are developed and made freely available by CNRS/INSU "Service National d'Observation"
Argo France (https://www.argo-france.fr/) at LOPS laboratory and at OSU IUEM (Univ. Brest). This work has received the
support of the French government within the framework of the "Investissements d'avenir" program integrated in France 2030
and managed by the Agence Nationale de la Recherche (ANR) under the reference "ANR-21-ESRE-0019". EP has been
funded by Natural Environment Research Council grant NE/W00755X/1. Argo data were collected and made freely
available by the International Argo Program and the national programs that contribute to it. The Argo Program is part of the
Global Ocean Observing System.
**Open Research** : Argo data are freely available at Argo (2000). ISAS T/S/O2 fields are freely available at Kolodziejczyk et
al. (2021). WOA18 fields and WOD18 are freely available at Boyer et al., (2018) and Boyer et al. (2016) , respectively.





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





*Figures*

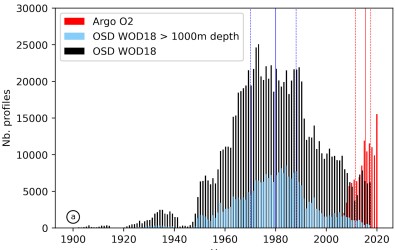

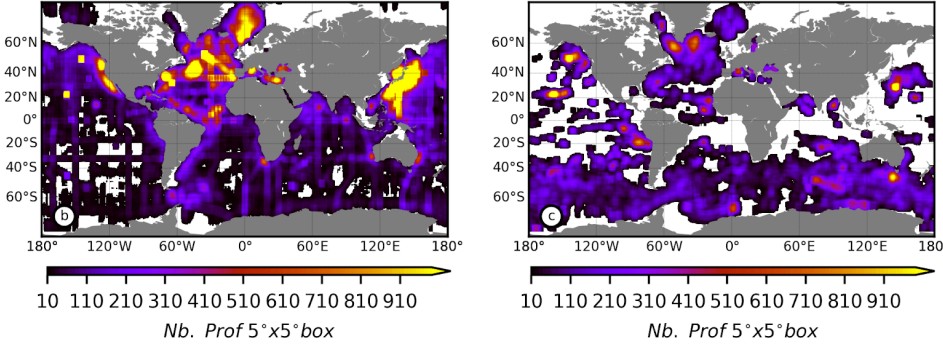

Figure 1 : a) Yearly distribution of DO profiles from WOD18 OSD used in WOA18 (black), WOD18 OSD profiles deeper than 1000 m depth (blue), and DMQC Argo dataset used in ISASO2 (red) over the period 1899-2019. Solid lines (dashed line) is the median (the first and third quartile) of the distribution. b) Density of DO OSD WOD18 profiles. c) Density of DO Argo profiles.



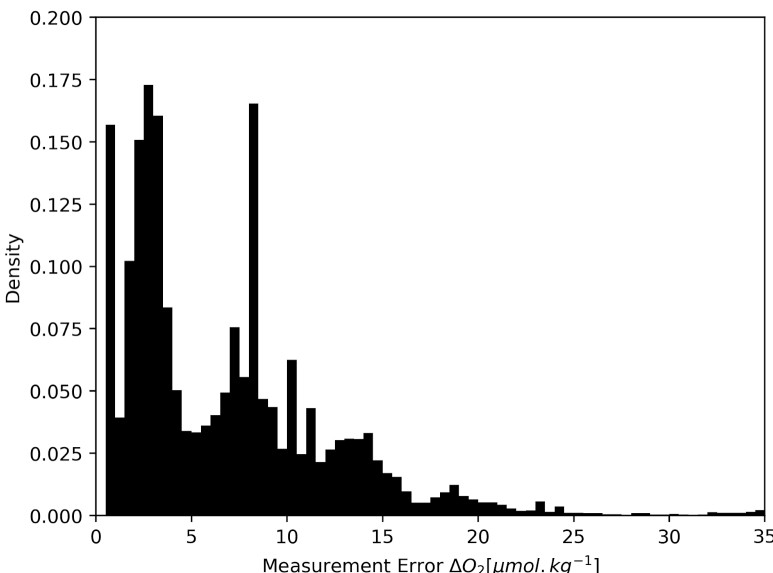


Figure 2: Distribution of the measurement error (DOXY_ADJUSTED_ERROR) from DO Argo S-profiles interpolated
on the ISAS standard z-levels.




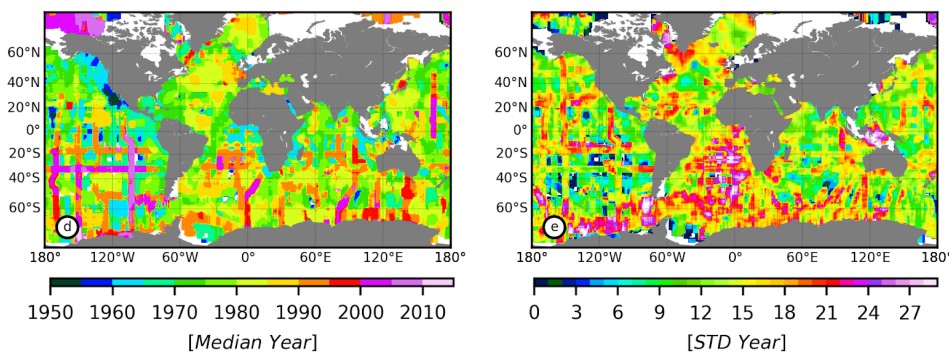


Figure 3 :d) Median year of sampling for DO OSD WOD18 profiles, and e) Standard deviation (in year) of the time

distribution for DO OSD WOD18 profiles, reaching at least 1000 m depth (in 5°x5° box).

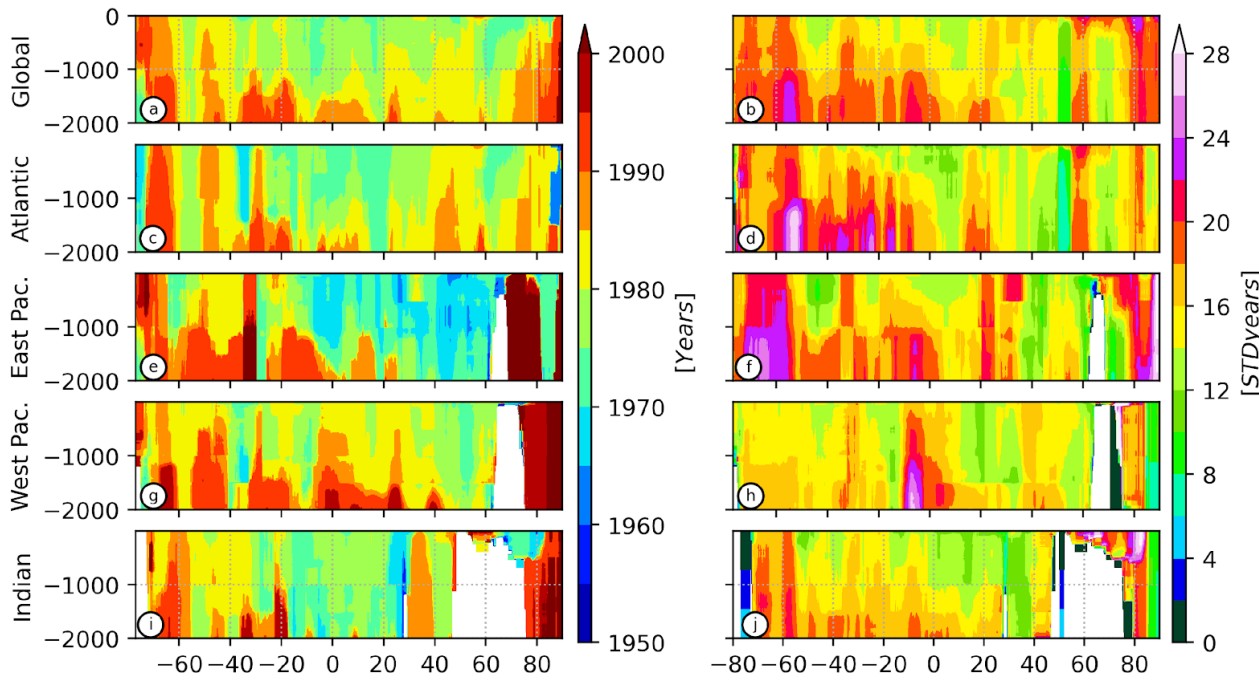


Figure 4: Meridional section of median time (left column) and STD (right column) for WOD18 OSD O2 profiles for
a,b) global ocean ; c,d) Atlantic basin (90°W-20°E) ; e,f) Eastern Pacific (180°W-90°W) ; g,h) Western Pacific
(120°E-180°E) and i,j) Indian (20°E-120°E). These time estimates are used to compute the equivalent trend for the
difference between WOA18 and ISAS_MEAN sections.






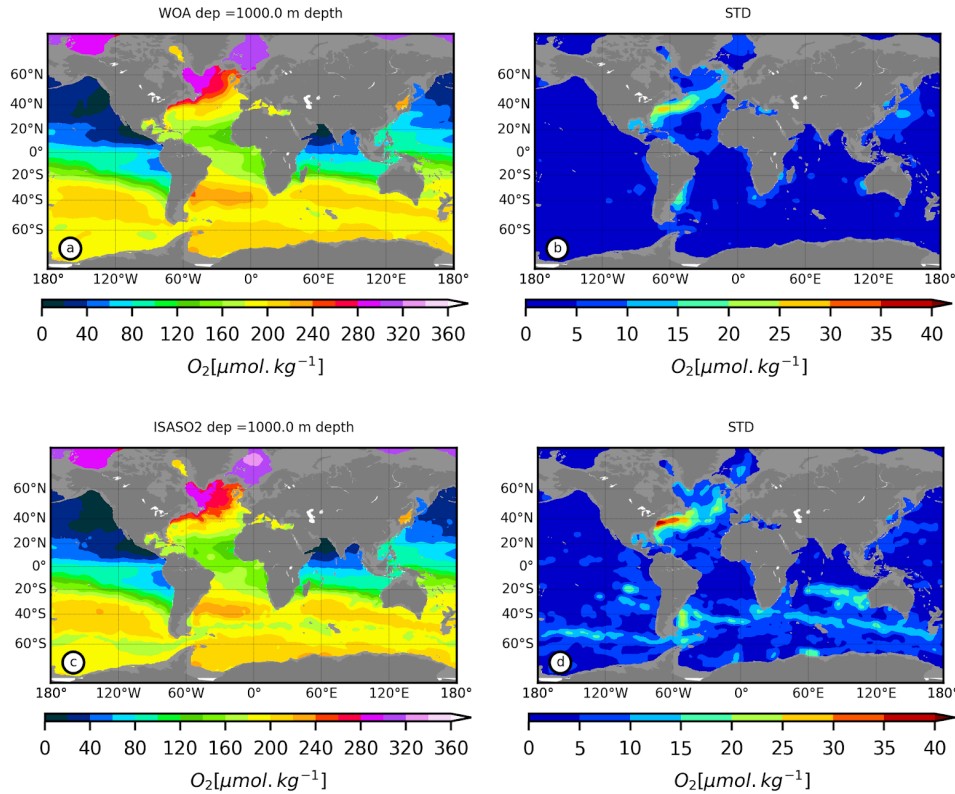


Figure 5: a) WOAO2 climatology of DO (in µmol.kg⁻¹) at 1000 m depth. b) WOAO2 climatology of STD (in µmol.kg⁻¹)
between 2005-2019 at 1000 m depth. c) ISASO2 climatology of DO (in µmol.kg⁻¹) between 2005-2019 at 1000 m depth. d)
ISASO2 climatology of STD (in µmol.kg⁻¹) between 2005-2019 at 1000 m depth.

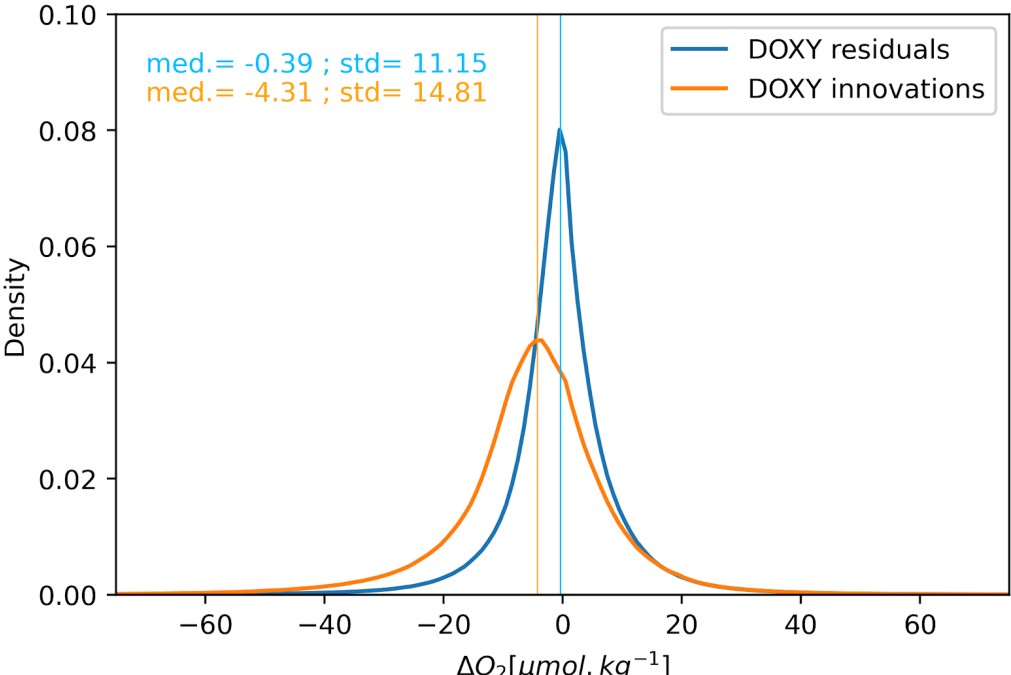


Figure 6: Density of residual distribution, i.e. difference between Argo data and ISAS_CLIM (blue) ; and innovation, i.e. difference between Argo data and first guess WOA18 (orange). The median for innovation is -4.31 µmol.kg$^{-1}$, suggesting a general deoxygenated bias between the WOA18 and Argo data likely due to i) a mean deoxygenation at the location of the DO Argo profile, or ii) remaining low oxygen bias in DO Argo profile. The median for residual is -0.39 µmol.kg$^{-1}$, suggesting that ISAS_CLIM estimates are slightly more oxygenated than DO Argo data at the location of the DO Argo profiles. This is likely due to the relaxation of the DO estimates toward the first guess (WOA18) and the smoothing in the OI procedure.

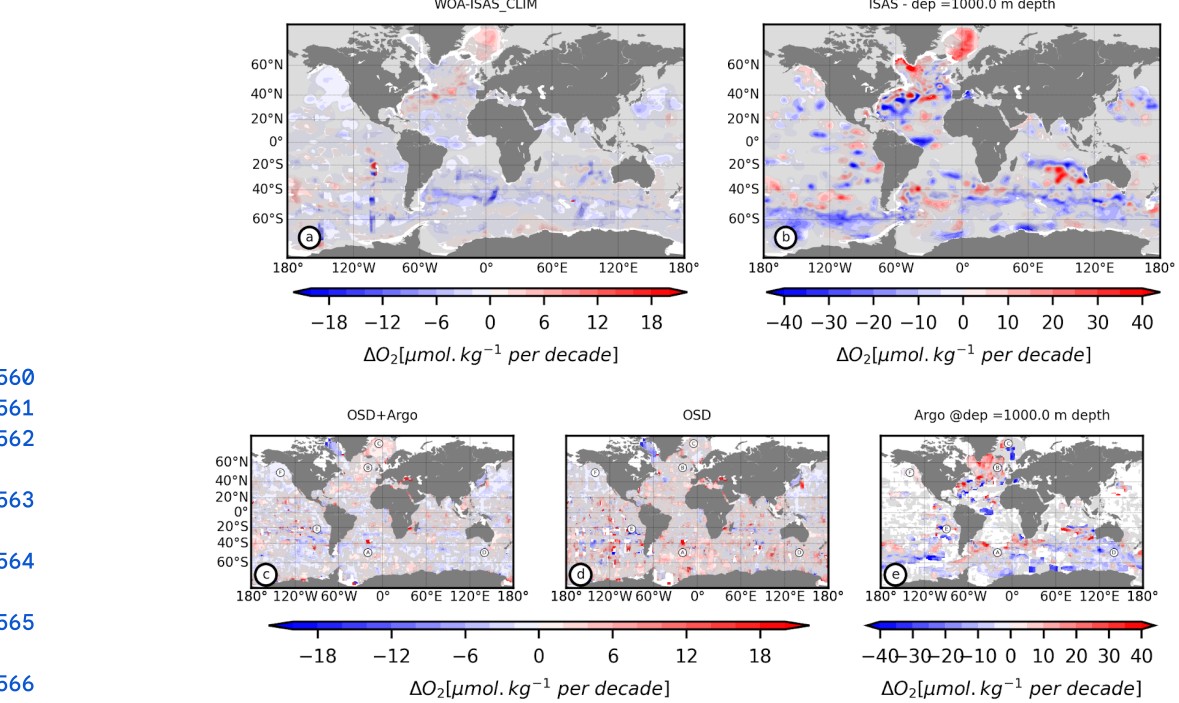

Figure 7: DO trend (in µmol.kg⁻¹ per decade) at 1000 m depth computed from a) difference between ISASO2_MEAN climatology minus WOA18 climatology, b) difference of the pendatal climatology (ISAS_M16 minus ISAS_M11); and the DO trend at 1000 m depth computed from individual DO profiles in 5°x5° boxes using c) OSD WOD18 + Argo data ; d) OSD WOD18 dataset alone ; and e) Argo DO dataset alone (2005-2020). Letter A-F on the cde) maps indicates the location of times series shown in Figure 8.



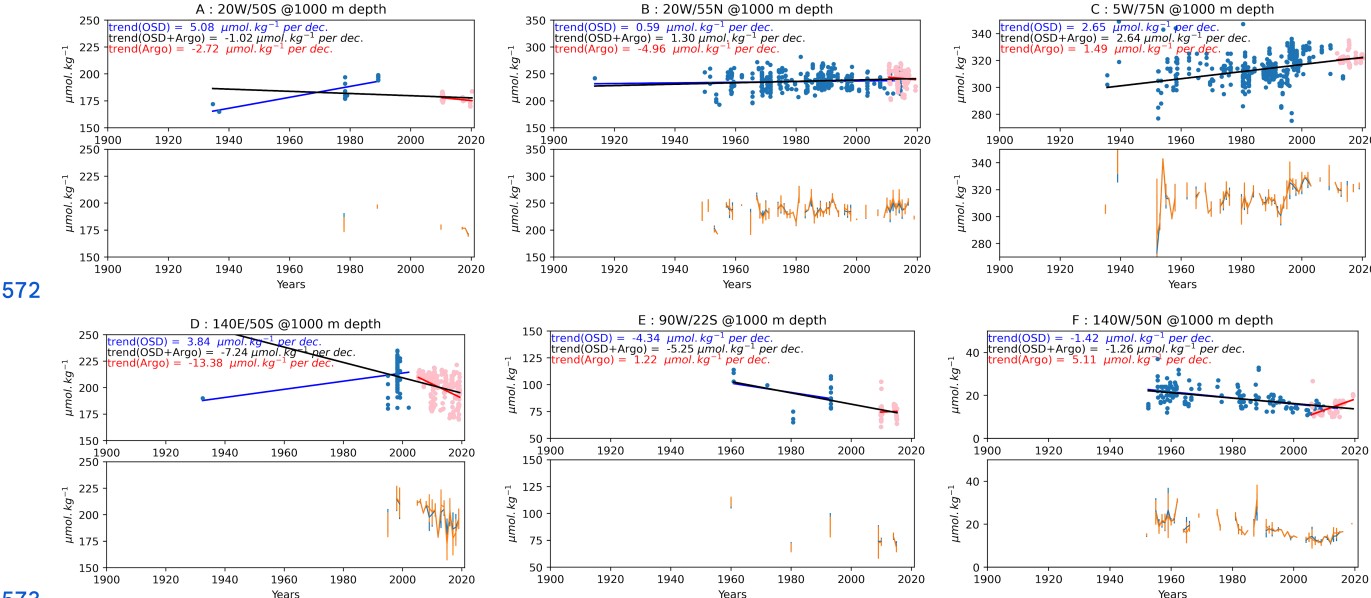



Figure 8: For each sub-panel: (upper sub-panel) DO trend computed in 5°x5° box at 1000 m depth for OSD WOD18 + Argo
(black) data ; OSD WOD18 dataset alone (blue) ; and  Argo DO dataset alone (2005-2020) (red). (lower sub-panel) : Time
series of yearly mean (blue) and median (orange) and associated STD using OSD WOD18 + Argo dataset. Each 5°x5° box
(see corresponding label on Fig. 7cbd) are centered on A : 20°W-50°S ; B :  20°W-55°N ; C : 5°W-75°N ; D : 20°W-140°E ;
E : 90°W-22°S and F : 140°W-50°N.

Figure 9:  a) WOA18-ISASO2_MEAN equivalent trends (mol.m$^{-2}$ per decade). Solid black (red) line are the WOA18

(ISASO2_MEAN) 40 and 80 µmol.kg$^{-1}$ contours. b) ISASO2_M16-ISASO2_M11 equivalent trends (mol.m$^{-2}$ per decade).

Equivalent trends smaller than uncertainties are grey shaded, and PCTVAR larger than 95% are blanked in Fig. 2a and b.



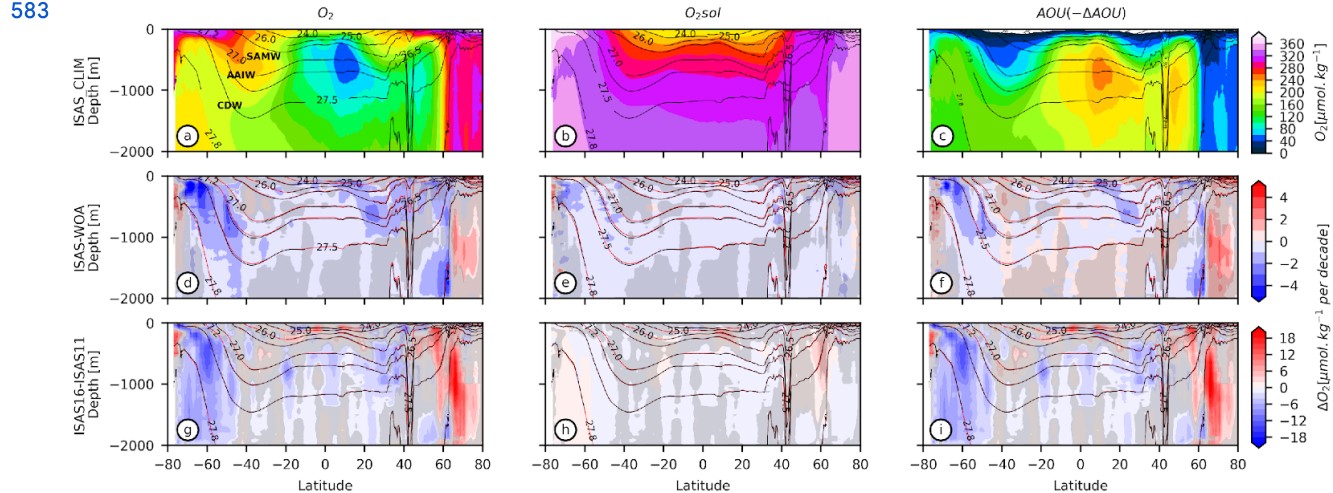

Figure 10: Global zonal average sections (ISASO2_MEAN), a) DO, b) Oxygen Saturation, c) Apparent Oxygen Utilization (AOU) (in µmol.kg$^{-1}$). ISASO2_CLIM-WOA18 d) DO difference, e) Oxygen Saturation anomaly, and f) -AOU anomaly. ISAS_M16-ISAS_M11 g) Dissolved Oxygen difference, h) Oxygen Saturation anomaly, and g) AOU anomaly. Black (red) contours correspond to the recent (older) climatological isopycnals (potential density in kg.m$^{-3}$). Shading indicates when equivalent trends are smaller than uncertainties.





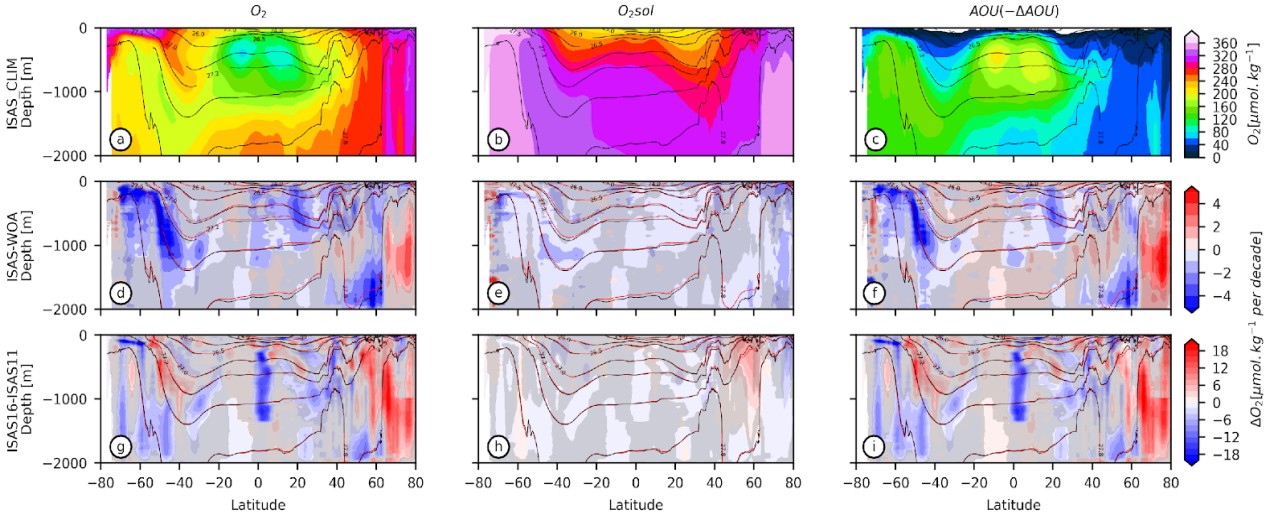


Figure 11: Same as Fig. 10 but for Atlantic basin only, averaged zonally (70°W-20°E).





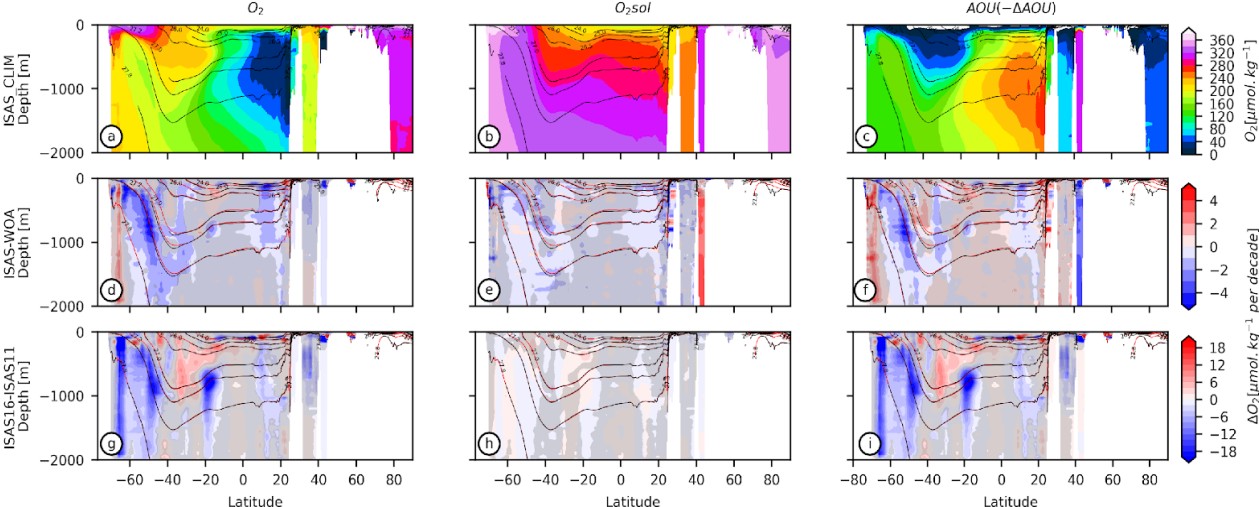


Figure 12: Same as Fig. 10 but for the Indian basin only, averaged zonally (20°E-120°E).



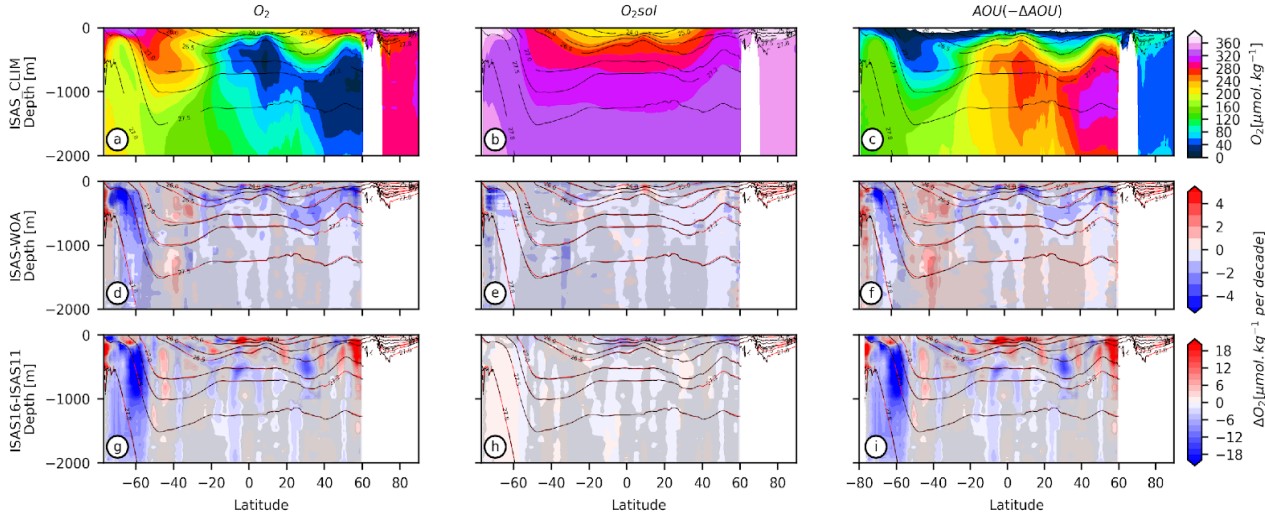


Figure 13: Same as Fig. 10 but for the Eastern Pacific section averaged zonally (180°E-70°W).




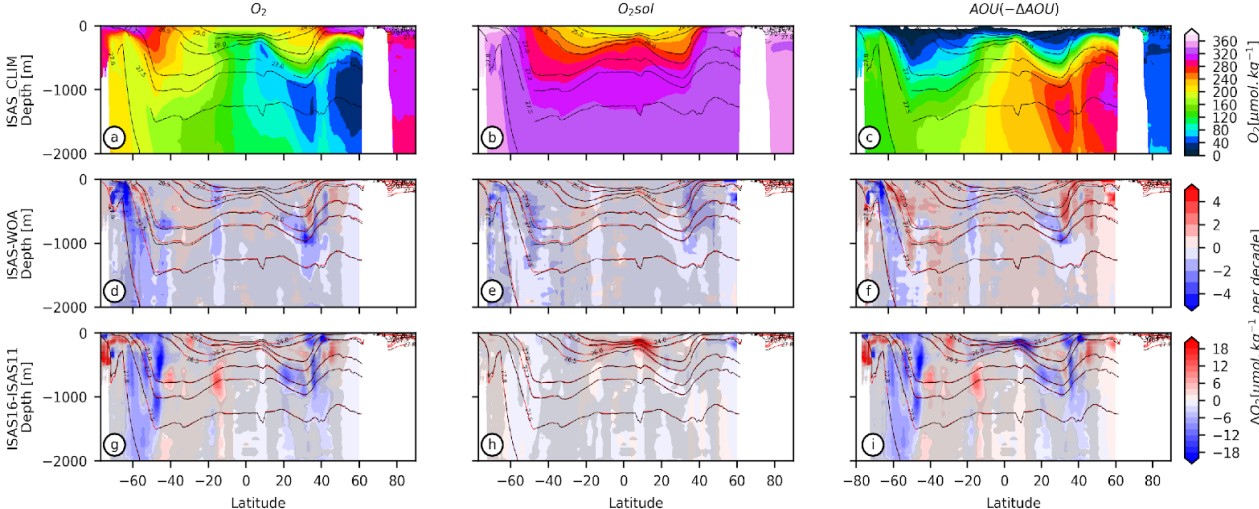


Figure 14: Same as Fig. 103 but for the Western Pacific section averaged zonally (120°E-180°E).