# Peer review of "ISASO2 : Recent trends and regional patterns of Ocean Dissolved Oxygen change"

_Earth System Science Data, 2024_

## Author Response (AR1)

**RC1**: 'Comment on essd-2024-106', Anonymous Referee #1, 13 May 2024 reply
Comment on "ISASO2 : Recent trends and regional patterns of Ocean Dissolved Oxygen change" by Nicolas Kolodziejczyk et al.

**General comments:**

This study focused on assessing changes in dissolved oxygen (DO) levels in the world's oceans over the last decades. Using a new dataset, ISASO2, derived from Argo-based optimally interpolated climatology, the study updates regional oxygen inventories and examines long-term and short-term deoxygenation trends. The paper is well written.

My primary concern is the need for a direct comparison with previously established gridded data, such as those by Schmidtko et al. (2017) and Sharp et al. (2023). It would be beneficial to emphasize how this new dataset might address some of the uncertainties previously noted in the field. Additionally, clarifying what distinguishes ISASO2 as potentially more reliable or different from earlier datasets could strengthen the impact and relevance of your findings.

*The authors thank the reviewer for their comments. We added the following text in the «Discussion » section :*

*(l.319-329)«In order to map regional DO concentration, ISASO2 takes advantage of the new Argo DO data, which resolves issues due to several sampling biases existing in the previous mappings such as Schmidtko et al. (2017). First, Argo DO have provided significant amounts of data in the Southern Hemisphere that were not available in the historical CTD dataset used in Schmidko et al. (2017). Second, while historical CTD have been carried out during summer season, Argo data resolves for the first time the seasonal cycle of DO in the sampled regions and better constrains DO uncertainties (Fig. 5).*

*        The GOBAI-O2 products (Sharp et al., 2023) has been generated using T,S,O2 Argo data and Artificial Intelligence algorithm to interpolate and extrapolate O2 data over the full Argo period. The advantage of this approach is providing longer timeseries and extrapolated profiles in poorly sampled regions. However, the uncertainties associated with such machine learning interpolation remain large, mainly in regions where the biological processes have an impact on O2 concentration, such as Oxygen Minimum Zones and need a larger amount of data to reduce it (Sharp et al., 2023). Here, with ISASO2, we choose an OI conventional approach that relies on few a priori hypotheses. Future deployment of Argo DO floats will help to improve sampling and the robustness of our estimates. »*

**Specific comments:**

Line 17: "developpement" should be "development"

*Done*

Line 65-66: Specify S-profiles and flag 1&2 to the general audience.

*This has been clarified in the revised manuscript : l.65-67 : «In this study, we used 117,359 Delayed Mode (DM) Argo profiles from the surface to 2000 m depth covering the period 2005-2019 (Fig.*

*1a). Only Quality Control (QC) flags set to 1 (good) and 2 (probably good) have been retained by default in the analysis process. »*

**RC2**: ['Comment on essd-2024-106'](), Anonymous Referee #2, 10 Jun 2024 reply
**Review: Recent trends and regional patterns of Ocean Dissolved Oxygen change**

It is very good to finally see some attention being paid to the promise of the ARGO based chemical climatology data set. This has been long promised, and is now much overdue. I commend the authors on their skill and diligence in ingesting all this information and on devising and implementing the techniques needed to extract information from this large and balky data set. I appreciate the hard work.

*The authors thank the reviewer for her/his supportive comments.*

BUT publication in this form would embarrass the authors, the Editor, and the Journal for it is rare to see an analysis that so completely misses the point of the study. The manuscript calculates the changes in the quantity of dissolved oxygen in the ocean seen as trends over time and comments on the losses due to the lower solubility of oxygen in warmer water. This is of no significance, although the calculations serve to beef up the observations. It is not clear in reading through this if care is taken to avoid including data in the euphotic zone where highly variable active oxygen production occurs – the brief mention that oxygen levels are increasing in the Nordic Seas seems to hint at that?

The concerns are for the impacts of changes in oxygen on marine life for food security, approaching anoxia, etc. The warm surface waters of the Mediterranean are just as hospitable to marine life as are the cold surface waters of the Nordic seas even though they contain less dissolved O2. The critical value is pO2, and a careful account is given in:

 **Hofmann, A.F.**, **E.T. Peltzer**, **P.M. Walz**, and **P.G. Brewer.** 2011. Hypoxia by degrees: Establishing definitions for a changing ocean. *Deep-Sea Research I,* 58: 1212–1226. http://dx.doi.org/10.1016/j.dsr.2011.09.004

It is true that absolute values are important too at very low oxygen concentrations where competition with NO3 ions comes into play:

**Brewer, P.G.**, A.F. Hofmann, **E.T. Peltzer**, and **W. Ussler III.** 2014. Evaluating microbial chemical choices: The ocean chemistry basis for the competition between use of O2 or NO3$^-$ as an electron acceptor. *Deep Sea Research I,* 87: 35–42. http://dx.doi.org/10.1016/j.dsr.2014.02.002

The original purpose of the ARGO program was strongly focused on observing the uptake of heat by the ocean and the penetration to depth of the present day warming signal. It would be of great value if the authors were to calculate the changes in pO2 – not just total oxygen – by combining the changing temperature and dissolved oxygen fields in a professional manner.

You may also wish to look at:

**Brewer, P.G.**, and **E.T. Peltzer.** 2017. Depth perception: the need to report ocean biogeochemical rates as functions of temperature, not depth. *Philosophical Transactions of the Royal Society A: Mathematical, Physical and Engineering Sciences,* 375(2102): 1–18. https://doi.org/10.1098/rsta.2016.0319

for understanding the critical role that temperature plays in controlling ocean oxygen consumption rates. Based on the well-developed skills I see displayed here this should be well within the capabilities of the authors.

*The authors thank the reviewer for pointing out this true fact that partial pressure of O2 matters for aerobic marine life sustainability (and the references that come along), more than the absolute dissolved oxygen concentration. While we agree on the need for such pO2 climatology for the marine life sustainability perspective, the goals of this manuscript are: i) to present a new interpolated oxygen dataset, and ii) to assess the change in the oxygen inventory in the ocean. Beyond the solubility of oxygen, the physical (i.e. mixing) and biogeochemical (i.e. consumption) processes contributing to AOU cannot be tackled in detail. Thus, we will leave for future work (and BGC specialists) This computation can be easily made, when needed, from the database we provide in this study, i.e., the complete set of P/T/S/O2 in ISAS and ISAS -O2 datasets.*

*We have modified the introduction and discussion of the manuscript to acknowledge this important point, as an application for the present dataset :*

*(l.30-37)"The latter has particular impact on the weak DO supply to the Oxygen Minimum Zones (OMZs), that are naturally low-oxygen, or hypoxic, regions located in the eastern tropical oceans (Karstensen et al., 2008 ; Paulmier and Ruiz-Pino, 2009, Hahn et al., 2017). One of the critical ecological impacts of the combined effect of global deoxygenation and ocean warming, is the decrease of the partial pressure of $O_2$ ($pO_2$ ; Hoffman et al., 2011 ; Brewer et al., 2014) and increase of biological oxygen consumption rate (Brewer and Peltzer, 2017). This is especially critical in the expending expansion of the OMZs (Stramma et al., 2008) with strong impact on the habitat of pelagic species (Stramma et al., 2012) and macrofaunal diversity (Sperling et al., 2016)."*

*(l.391-395)"Therefore, long-term observations and consistent spatial coverage of the ocean DO, combined with other P/T/S and BGC parameters in the framework of OneArgo will provide further insight on the regional mechanisms of ocean deoxygenation, such as thermodynamic and chemical driver of biological consumption (e.g. Hoffman et al., 2011 ; Brewer et al., 2014 ; Brewer and Peltzer, 2017)."*

---

## Author Response (AR2)

**Response to editor to minor revision request:**

**Public justification (visible to the public if the article is accepted and published)**:

Dear authors,

I have now reviewed the referee reports and your response to them.

I am overall happy with your revisions, but still want to request some changes.

Referee 2 has highlighted issues with the use of data in the euphotic zone to assess trends, as well as the use of concentrations instead of partial pressure, that have not been fully addressed in the revisions. Because ESSD is a journal that focusses on the data rather than the interpretation, I would request you to tune down the extensive discussion and interpretation of the patterns, and rather focus on the potential limitations (such as the euphotic zone issue), uncertainty, and comparison with the older datasets. This will also make the concerns of Referee 2 redundant for this manuscript.

This does not require a main overhaul, rather a reduction of paragraphs, mainly L272-287, L299-307, L335-373. Try to avoid extensively discussing mechanisms and drivers, and focus on the patterns your dataset reveals.

Kind regards

Sebastiaan

*The authors thank the reviewers and editors for pointing out remaining issues in the manuscript. We have carefully followed the advices and reduce the extensive discussions that are not directly related to the description and validation of the observed pattern in ISASO2. We especially reduce the results and discussion paragraphs at l.298, l. 335, l. 355, l. 367 in the revised version of the manuscript (see the manuscript with tracked changes).*

*We also add a sentence to acknowledge the uncertainty in the DO inventory trends due to including euphotic zone in our computation (l. 337): "It should be noted that DO inventory over the full water column includes the euphotic zone, that may be highly variable in DO, and part of the variability at depth (below 2000 m depth) is overlooked during the Argo period."*

*Eventually, we hope that the manuscript is known suitable for publication in ESSD.*